# Exposure to live saprophytic *Leptospira* before challenge with a pathogenic serovar prevents severe leptospirosis and promotes kidney homeostasis

**Suman Kundu†, Advait Shetty†‡, Maria Gomes-Solecki\***

Department of Microbiology, Immunology and Biochemistry, University of Tennessee Health Science Center, Memphis, United States

## eLife assessment

This **important** study contributes to our understanding on how prior exposure to a non-pathogenic Leptospira strain could prime the host to prevent severe leptospirosis following infection with a pathogenic strain. The work described is **solid** and broadly supports the claims, with minor weaknesses that could be addressed in future studies. The work will be of interest to scientists interested in host-pathogen interactions and leptospirosis.

**\*For correspondence:**
mgomesso@uthsc.edu

†These authors contributed equally to this work

**Present address:** ‡MLM Medical Labs Memphis, Memphis, United States

**Abstract** Previous studies demonstrated that *Leptospira biflexa*, a saprophytic species, triggers innate immune responses in the host during early infection. This raised the question of whether these responses could suppress a subsequent challenge with pathogenic *Leptospira*. We inoculated C3H/HeJ mice with a single or a double dose of *L. biflexa* before challenge with a pathogenic serovar, *Leptospira interrogans* serovar Copenhageni FioCruz (LIC). Pre-challenge exposure to *L. biflexa* did not prevent LIC dissemination and colonization of the kidney. However, it rescued weight loss and mouse survival thereby mitigating disease severity. Unexpectedly, there was correlation between rescue of overall health (weight gain, higher survival, lower kidney fibrosis marker ColA1) and higher shedding of LIC in urine. This stood in contrast to the *L. biflexa* unexposed LIC challenged control. Immune responses were dominated by increased frequency of effector T helper (CD4+) cells in spleen, as well as significant increases in serologic IgG2a. Our findings suggest that exposure to live saprophytic *Leptospira* primes the host to develop Th1 biased immune responses that prevent severe disease induced by a subsequent challenge with a pathogenic species. Thus, mice exposed to live saprophytic *Leptospira* before facing a pathogenic serovar may withstand infection with far better outcomes. Furthermore, a status of homeostasis may have been reached after kidney colonization that helps LIC complete its enzootic cycle.

## Introduction

Leptospirosis, a neglected re-emerging enzootic spirochetal disease, affects millions of people worldwide causing an overall mortality rate of 65,000 per year (**Costa et al., 2015**). In addition, it causes serious health problems in animals of agricultural interest which leads to substantial economic losses mostly in tropical and subtropical countries. Assessing the true severity of leptospirosis can be incredibly challenging, especially when early diagnosis is difficult due to nonspecific symptoms that overlap with other illnesses (**Haake and Levett, 2015**). Recent outbreaks of both human and canine leptospirosis in New York and California in 2020–2022 (**NYC Health, 2021**; **Health, 2022**) underscores

the need for development of effective strategies to control this disease. Although serovar-specific vaccines are available for animals and at least one is available for humans, no broadly effective vaccine is available for either (*Barazzone et al., 2021*). The absence of an effective cross-protective vaccine candidate increases the risk of disease re-emergence on a global scale. Efforts to use leptospiral surface antigens in various vaccine formulations have shown limited success in conferring protection against leptospiral dissemination and shedding, as well as severe disease. *Leptospira* immune evasion strategies contribute to the complexities of finding good vaccine candidates.

The genus *Leptospira* is broadly categorized into two major clades P and S (P̲athogens and S̲apro-phytes) and further categorized into four subclades, P1, P2, S1, and S2 based on their virulence properties, growth conditions, and genetic make-up (*Vincent et al., 2019*; *Picardeau, 2017*). Subclade P1 is further divided in two phylogenetically related groups named P1+ (high-virulence pathogens, established pathogenic species, e.g. *Leptospira interrogans*) and P1- (low-virulence pathogens, phenotypically not well characterized) (*Giraud-Gatineau et al., 2024*). *Leptospira* survives in moist conditions and are free-living organisms naturally found in soil and water (*Narkkul et al., 2020*; *Benacer et al., 2013*). The spread of infection occurs through contaminated water contact with breached skin or mucosal surfaces (*Ko et al., 2009*). Saprophytic strains of *Leptospira,* such as *Leptospira biflexa* (S1), are unable to establish disease due to the lack of certain virulence factors (*Picardeau, 2017*) and have been found in natural environments around the world alongside pathogenic serovars (*Vincent et al., 2019*; *Ko et al., 2009*; *Guglielmini et al., 2019*). Moreover, *L. biflexa* exhibits certain niche-specific adaptations that allow them to persist in both environmental and host settings (*Zhang et al., 2018*; *Castiblanco-Valencia et al., 2016*).

Our previous studies (*Shetty et al., 2021*; *Kundu et al., 2022*) demonstrated that *L. biflexa* triggers a robust innate immune response in mice during the acute phase of infection. This raised the question of whether saprophytic *Leptospira*-induced immune responses could confer any degree of resistance or immune memory that could suppress a subsequent challenge with a pathogenic serovar of *Leptospira*. Answering that question was the main goal of the current study.

## Results

### Exposure to saprophytic *Leptospira* before infection with a pathogenic serovar prevents disease and increases survival of C3H-HeJ mice

We inoculated adult C3H-HeJ male mice with a single dose of *L. biflexa* 2 weeks before challenge with *L. interrogans* (LB[1]LIC[1]) at 8 weeks (*Figure 1A*) and measured a significant rescue of weight loss over a period of 15 days as compared to mice infected at 8 weeks that did not receive *L. biflexa* (PBS[1]LIC[1]) (p<0.0001); unchallenged control mice that received *L. biflexa* (LB[1]) or PBS (PBS[1]) gained weight throughout the corresponding 15 days (*Figure 1B*). Survival curves were generated after the mice reached the following endpoint criteria: 20% weight loss or 15 days post challenge with *L. interrogans* or 15 days post inoculation with *L. biflexa*/PBS for the controls (*Figure 1C*). All mice infected with *L. interrogans* (PBS[1]LIC[1]) reached the 20% weight loss endpoint criteria between d9 and d12 post infection. In contrast, 75% of the mice that received one dose of *L. biflexa* before challenge with *L. interrogans* (LB[1]LIC[1]) survived and gained significant body weight (*Figure 1B*) which was similar to the naïve control that received only PBS. Analysis of bacterial dissemination was done by qPCR of the *Leptospira* 16S gene in genomic DNA purified from blood, kidney tissue, and urine. Of note, although 16S rRNA primers can amplify *L. biflexa* 6 hr post infection (*Surdel et al., 2022*), we processed the tissue samples 30 days or 45 days post *L. biflexa* exposure. We also found that a single exposure to *L. biflexa* before challenge did not prevent dissemination of pathogenic *L. interrogans* in blood (*Figure 1—figure supplement 1*) or shedding in urine (*Figure 1D*), or kidney colonization (*Figure 1E*). Culture of kidney in EMJH media showed presence of ~2500 motile, morphologically intact *L. interrogans* under dark-field microscopy which was confirmed by 16S qPCR (*Figure 1F*) both on d3 and d5 post culture of kidney collected from LB[1]LIC[1] mice; kidney from PBS[1], LB[1], and PBS[1]LIC[1] did not produce positive cultures by dark-field microscopy or 16S qPCR (*Figure 1—source data 1*, *Figure 1—figure supplement 1—source data 1*).

In the double exposure study, mice were inoculated with two bi-weekly doses of *L. biflexa* 2 weeks before challenge with *L. interrogans* (LB[2]LIC[2]) at 10 weeks in comparison with the respective controls (*Figure 2A*). As expected, mice infected at 10 weeks with LIC that did not receive *L. biflexa* (PBS[2]LIC[2])

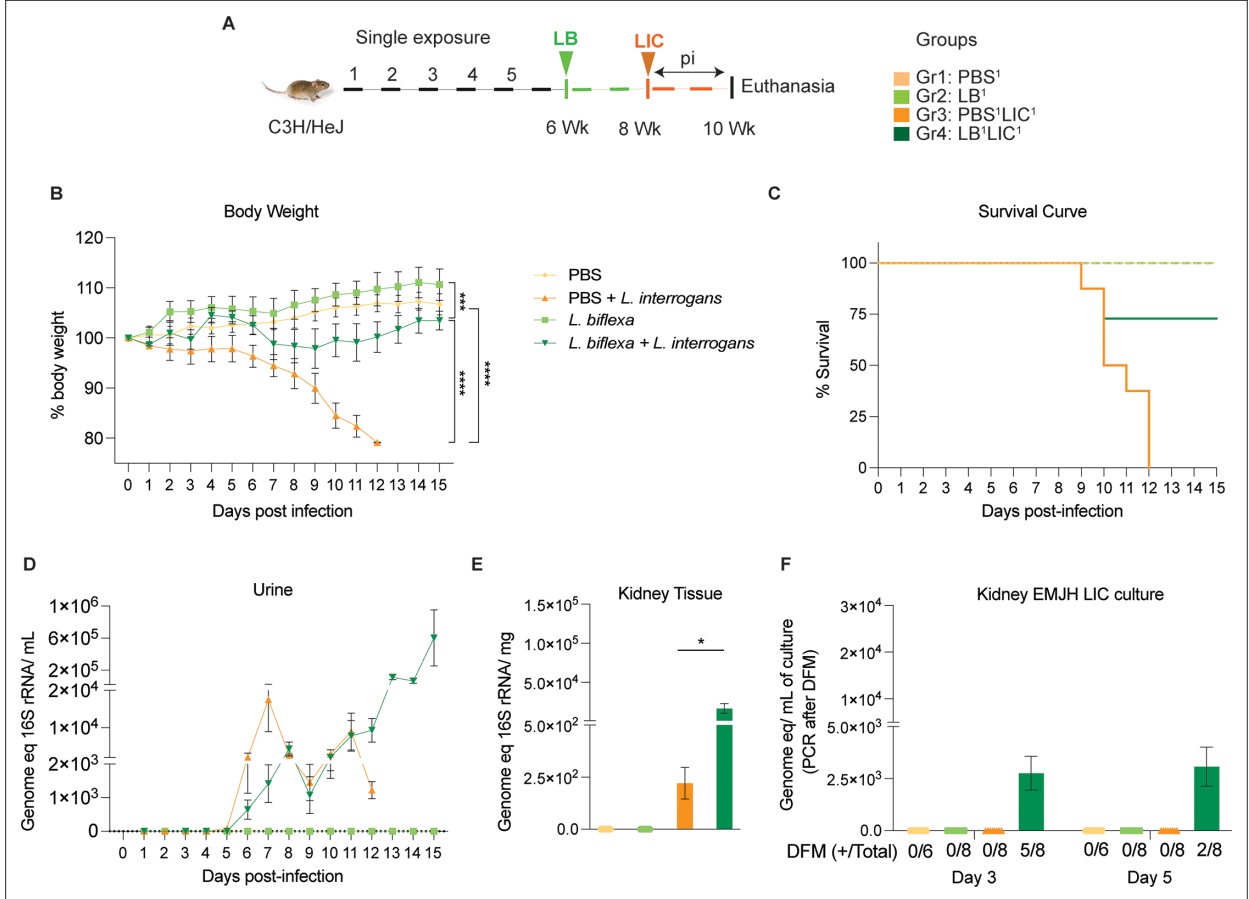

**Figure 1.** Weight loss, kidney colonization, shedding in urine, and survival to challenge with *L. interrogans* following a single exposure to *L. biflexa*. Male C3H/HeJ mice were inoculated once with $10^8$ *L. biflexa* (LB) at 6 weeks and they were challenged with $10^8$ *L. interrogans* serovar Copenhageni FioCruz (LIC) at 8 weeks. (**A**) Experimental layout; (**B**) body weight measurements (%) acquired for 15 days post challenge with LIC; (**C**) mouse survival within the 15 days post challenge with LIC; (**D**) 16S rRNA qPCR quantification of live LIC in urine; (**E**) 16S rRNA qPCR quantification of *Leptospira* burden in kidney tissue harvested on d15 post challenge with LIC and (**F**) 16S rRNA qPCR from kidney EMJH cultures containing live *Leptospira* previously observed by dark-field microscopy (DFM). DFM positive culture from the total data is represented in numbers under the graph. Statistical analysis was performed by ordinary one-way ANOVA followed by Tukey's multiple comparison correction between challenged groups and their respective controls, *p<0.05, **p<0.01, ***p<0.001, and ****p<0.0001, N=6–8 mice per group. Data represents two independent experiments.

The online version of this article includes the following source data and figure supplement(s) for figure 1:

**Source data 1.** Excel file containing the source data used to make *Figure 1*.

**Figure supplement 1.** qPCR to quantify *L. interrogans* load in blood of mice using 16S rRNA *Leptospira*-specific primers and probes from the single *L. biflexa* exposure experiment.

**Figure supplement 1—source data 1.** Excel file containing the data used to make *Figure 1—figure supplement 1*.

lost ~11% of weight on d11 post challenge and did not recover (*Figure 2B*). In contrast, mice that received a double dose of *L. biflexa* 2 weeks before challenge at 10 weeks with *L. interrogans* ($LB^2LIC^2$) lost a maximum of 5% weight on d10 and recovered fully by d15 post infection; unchallenged control mice that received *L. biflexa* ($LB^2$) or PBS ($PBS^2$) gained weight throughout the 15 days (*Figure 2B*). Survival curves generated after the mice reached endpoint criteria (*Figure 2C*) show that all experimental groups survived LIC infection. Analysis of bacterial dissemination showed that a double exposure to *L. biflexa* before challenge did not prevent dissemination of pathogenic *L. interrogans* in blood (*Figure 2—figure supplement 1*), or shedding in urine (*Figure 2D*), or kidney colonization (*Figure 2E*). Culture of kidney in EMJH media showed presence of 5000–10,000 motile, morphologically intact *L. interrogans* on d3 and d5 post culture of kidney collected from $PBS^2LIC^2$ mice in contrast to 1000–2500 live *L. interrogans* observed in culture from kidney collected from $LB^2LIC^2$ mice which was confirmed by 16S qPCR (*Figure 2F*); kidney from $PBS^2$ and $LB^2$ mice did not produce positive

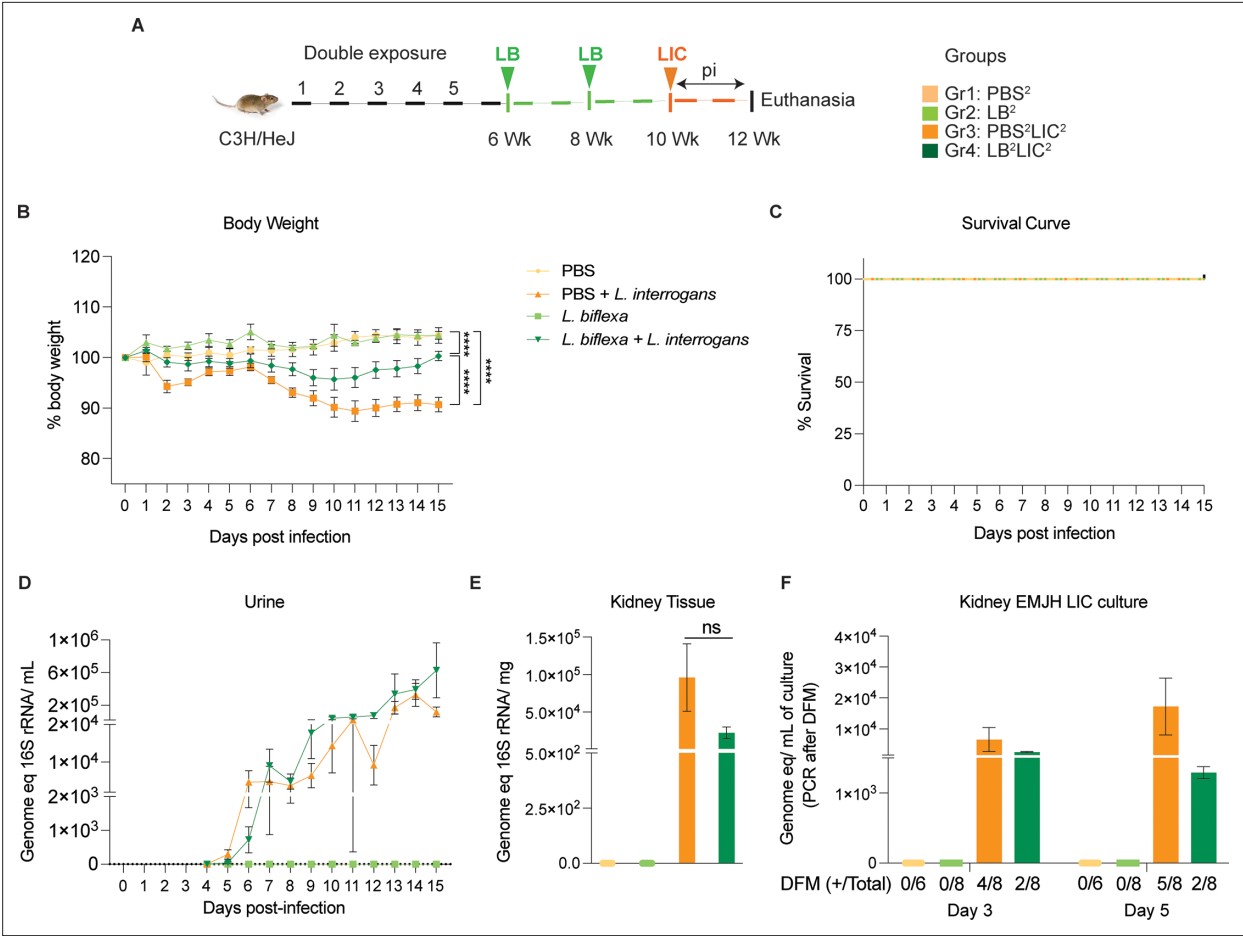

**Figure 2.** Weight loss, kidney colonization, shedding in urine, and survival to challenge with *L. interrogans* following a double exposure to *L. biflexa*. Male C3H/HeJ mice were inoculated twice with $10^8$ *L. biflexa* at 6 and 8 weeks, and at 10 weeks they were challenged with $10^8$ *L. interrogans* ser Copenhageni FioCruz (LIC). (**A**) Experimental layout; (**B**) body weight measurements (%) acquired for 15 days post challenge with LIC; (**C**) mouse survival within the 15 days post challenge with LIC; (**D**) 16S rRNA qPCR quantification of live LIC in urine; (**E**) 16S rRNA qPCR quantification of *Leptospira* burden in kidney tissue harvested on d15 post challenge with LIC and (**F**) 16S rRNA qPCR from kidney EMJH cultures containing live *Leptospira* previously observed by dark-field microscopy (DFM). DFM positive culture from the total data is represented in numbers under the graph. Statistical analysis was performed by ordinary one-way ANOVA followed by Tukey's multiple comparison correction between challenged groups and their respective controls, *p<0.05, **p<0.01, ***p<0.001, and ****p<0.0001. N=6–8 mice per group. Data represents two independent experiments.

The online version of this article includes the following source data and figure supplement(s) for figure 2:

**Source data 1.** Excel file containing the data used to make *Figure 2*.

**Figure supplement 1.** qPCR to quantify *L. interrogans* load in blood of mice using 16S rRNA *Leptospira*-specific primers and probes from the double *L. biflexa* exposure experiment. Data represents two experiments.

**Figure supplement 1—source data 1.** Excel file containing the data used to make *Figure 2—figure supplement 1*.

---

cultures by dark-field microscopy or 16S qPCR (*Figure 2—source data 1*, *Figure 2—figure supplement 1—source data 1*).

### *L. biflexa* exposure before challenge with *L. interrogans* mitigates renal histopathological changes

As expected, hematoxylin and eosin (H&E) staining of histological slices of all kidneys from mice challenged with *L. interrogans* in both single and double exposure experiments (PBS[1]LIC[1] and PBS[2]LIC[2]) showed signs of inflammation with increased immune cell infiltration (*Figure 3A and C*). In contrast, H&E staining of kidney slices from the groups of mice exposed to *L. biflexa* before *L. interrogans* challenge (LB[1]LIC[1] and LB[2]LIC[2]) showed reduced immune cell infiltration. We also measured expression of a marker (ColA1) for kidney fibrosis. In both experiments, kidneys from mice challenged with

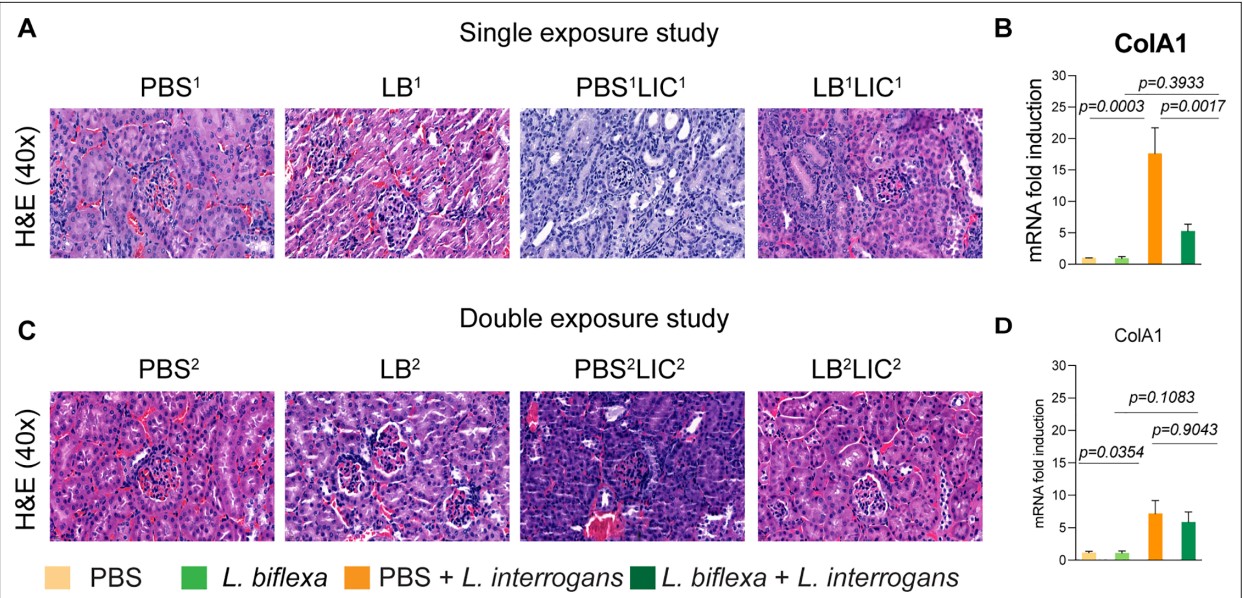

**Figure 3.** Kidney histopathology and quantification of renal fibrosis. Representative hematoxylin and eosin (H&E)-stained kidney tissue sections from both single and double exposure studies are included in (**A**) and (**C**), respectively. The images were captured at ×40 magnification. (**B**) and (**D**) represent the mRNA expression of kidney fibrosis marker ColA1 by qPCR normalized to endogenous β-actin expression. Data was analyzed by ordinary one-way ANOVA followed by Tukey's multiple comparison correction between challenged groups with their respective controls; *p-values are included in the graphs. Data represents one of two independent experiments.

The online version of this article includes the following source data and figure supplement(s) for figure 3:

**Source data 1.** Excel file containing the data used to make *Figure 3*.

**Figure supplement 1.** Morphometric analysis of kidney of mice from the double *L. biflexa* exposure experiment. Data represents one experiment.

*L. interrogans* (PBS[1]LIC[1] and PBS[2]LIC[2]) had significantly higher expression of ColA1 as compared to the controls; in contrast, kidneys from mice challenged with *L. interrogans* after exposure to *L. biflexa* (LB[1]LIC[1] and LB[2]LIC[2]) were not different than the controls (*Figure 3B and D*; *Figure 3—source data 1*).

In addition, we were able to collect kidneys from experimental mice subjected to the double exposure of *L. biflexa* because they all survived subsequent challenge with *L. interrogans*. As such, we did a comparative gross morphological analysis between the four groups (*Figure 3—figure supplement 1*). We found that kidney from LB[2]LIC[2] mice maintained their normal gross anatomy and coloration as did kidneys from control mice (PBS[2] and LB[2]), in contrast to kidneys from mice challenged with *L. interrogans* that were not previously exposed to *L. biflexa* (PBS[2]LIC[2]).

## Serologic IgG2a responses to *L. interrogans* were significantly higher in mice pre-exposed to *L. biflexa* before challenge with *L. interrogans*

In both single and double *L. biflexa* exposure experiments we measured anti-*L. interrogans* total IgM, total IgG, and IgG subtypes IgG1, IgG2a, and IgG3 in serum collected 2 weeks after challenge with *L. interrogans* (IgG subtypes shown in *Figure 4A and B*). In both experiments, total IgM and IgG were significantly increased in PBS-LIC and LB-LIC when compared to the respective controls, but not between PBS-LIC and LB-LIC groups. Regarding IgG isotypes, IgG1 was generally low and IgG2a as well as IgG3 were generally high in groups infected with *L. interrogans*. Although differences between groups (PBS[2]v PBS[2]LIC[2] and LB[1/2] vs LB[1/2]LIC[1/2]) were significant, differences between the LIC infected groups (PBS[1/2]LIC[1/2] vs LB[1/2]LIC[1/2]) were not significant for IgG1 and for IgG3 in contrast to IgG2a (p=0.001 for single exposure and p=0.0095 for double exposure) (*Figure 4—source data 1*).

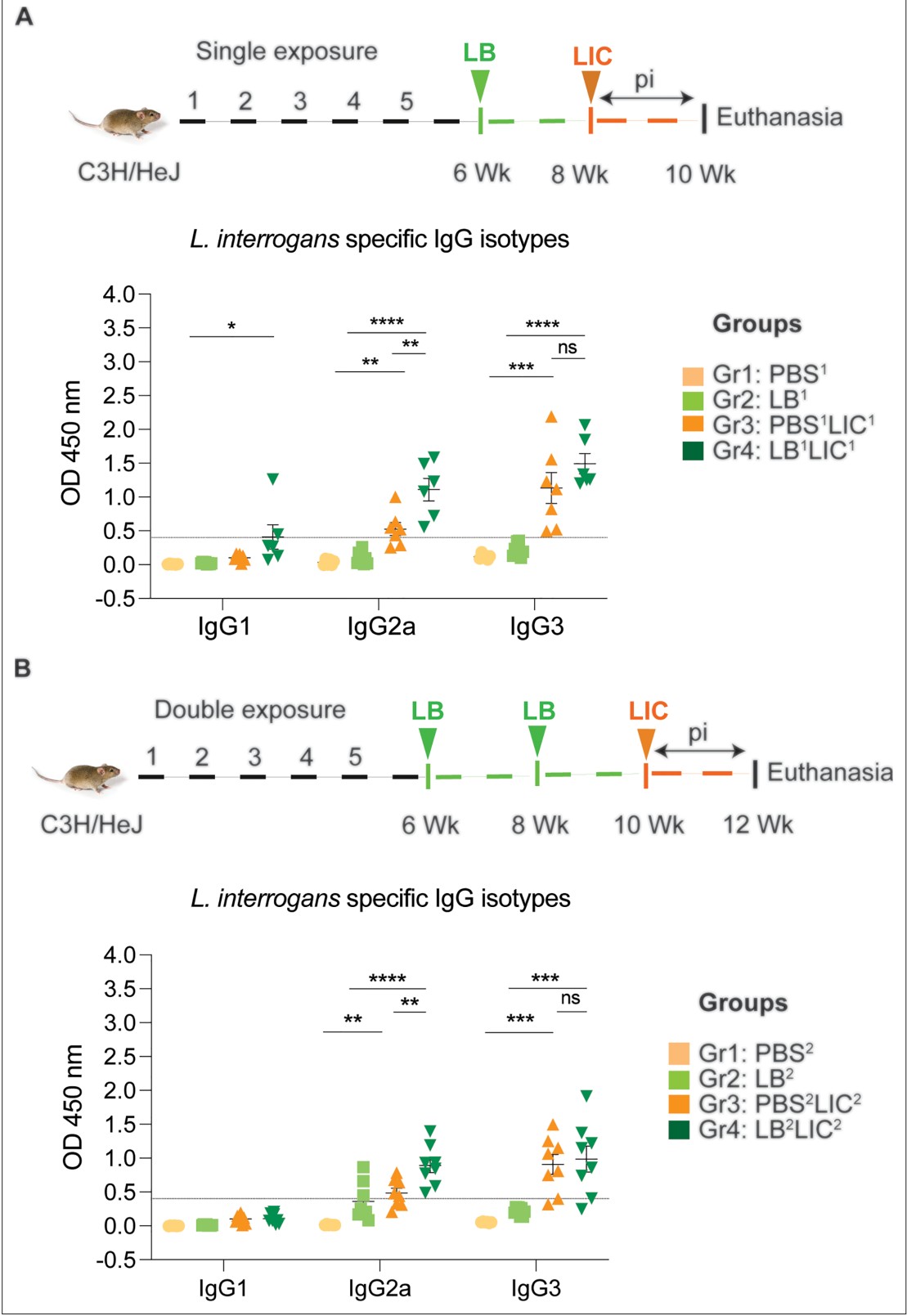

**Figure 4.** Detection of IgG1, IgG2a, and IgG3 specific to *L. interrogans* in serum from experimental mice. (**A**) represents IgG isotypes specific to *L. interrogans* in 10-week serum of mice exposed once to *L. biflexa* before *L. interrogans* challenge. (**B**) represents IgG isotypes specific to *L. interrogans* in 12-week serum of mice exposed twice to *L. biflexa* before *L. interrogans* challenge. Ordinary one-way ANOVA followed by Tukey's multiple comparison

*Figure 4 continued on next page*

*Figure 4 continued*

correction test was used to compare between challenged groups with their respective controls; *p<0.05, **p<0.01, ***p<0.001, ****p<0.0001, and ns = not significant; N=6–8 mice per group. Data represents two independent experiments.

The online version of this article includes the following source data for figure 4:

**Source data 1.** Excel file containing the raw data points used to make *Figure 4*.

## Exposure to non-pathogenic *L. biflexa* before pathogenic *L. interrogans* challenge induced increased frequencies of effector helper T cells in spleen

We immunophenotyped the spleen cells from the mice subjected to double *L. biflexa* exposure because all animals survived to the term of the experiment. The gating strategy used for spleen cell immunophenotyping is provided in *Figure 5—figure supplement 1*. Mice in the single exposure experiment met endpoint criteria before the term of the experiment and thus we were not able to process spleen for immunophenotyping. In the *L. biflexa* double exposure experiment, we measured increased frequencies in B cells when LIC infected mice were compared to the respective controls, but not between PBS$^2$LIC$^2$ and LB$^2$LIC$^2$ mice (*Figure 5A*). We measured decreased frequencies in T cells when LIC infected mice were compared to the respective controls but not between PBS$^2$LIC$^2$ and LB$^2$LIC$^2$ mice (*Figure 5B*). No differences were observed in NK cells between any of the groups (*Figure 5C*). We measured increased frequencies in helper T cells between all groups; of note, PBS$^2$LIC$^2$ vs LB$^2$LIC$^2$ p=0.006 (*Figure 5D*). We also measured decreased frequencies in cytotoxic T cells between all groups; of note PBS$^2$LIC$^2$ vs LB$^2$LIC$^2$ p=0.0056 (*Figure 5E*; *Figure 5—source data 1*).

Furthermore, T cell subset typing (*Figure 6*) showed that frequency of early effector CD4+ T helper cells (*Figure 6B*, CD44-CD62L-) and effector T helper cells (*Figure 6C*, CD44+CD62L-) were significantly increased when compared between the LIC challenged groups and the respective controls (except PBS$^2$ vs PBS$^2$LIC$^2$ early effectors) and that frequency of early effector and effector T helper cells was higher in the LB$^2$LIC$^2$ group than PBS$^2$LIC$^2$. No major changes were measured in memory CD4+ T helper cells (*Figure 6D*, CD44+CD62L+). In the CD8+ cytotoxic T cell subsets, we measured significant decreases in frequency of naïve T cells between LIC infected groups and the respective controls (*Figure 6E*, CD62L+CD44-) and this was replicated in the CD8+ cytotoxic memory except

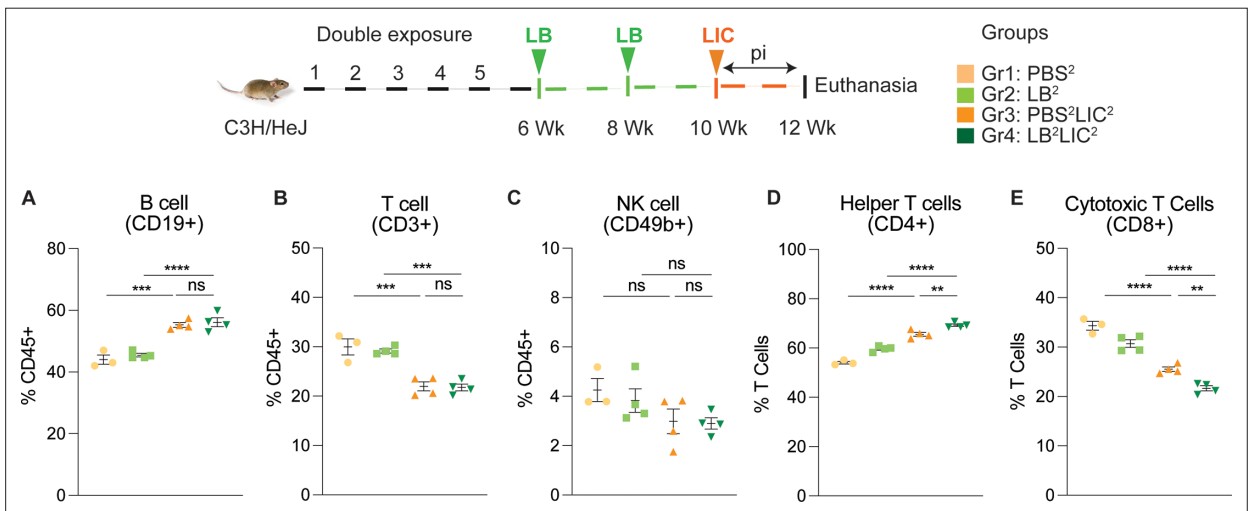

**Figure 5.** Frequency of lymphocytes in spleen of mice subjected to a double exposure of *L. biflexa* before challenge with *L. interrogans*. (**A–E**) show B cell (CD19+), T cell (CD3+), NK cell (CD49b+), helper T cell (CD4+), and cytotoxic T cell (CD8+) frequencies in groups of experimental mice. Ordinary one-way ANOVA followed by Tukey's multiple comparison correction test was used to compare between challenged groups and their respective controls; **p<0.01, ***p<0.001, ****p<0.0001, and ns = not significant; N=3–4 mice per group. Data represents one of two independent experiments.

The online version of this article includes the following source data and figure supplement(s) for figure 5:

**Source data 1.** Excel file containing data points used to generate *Figure 5*.

**Figure supplement 1.** Flow cytometry gating strategy.

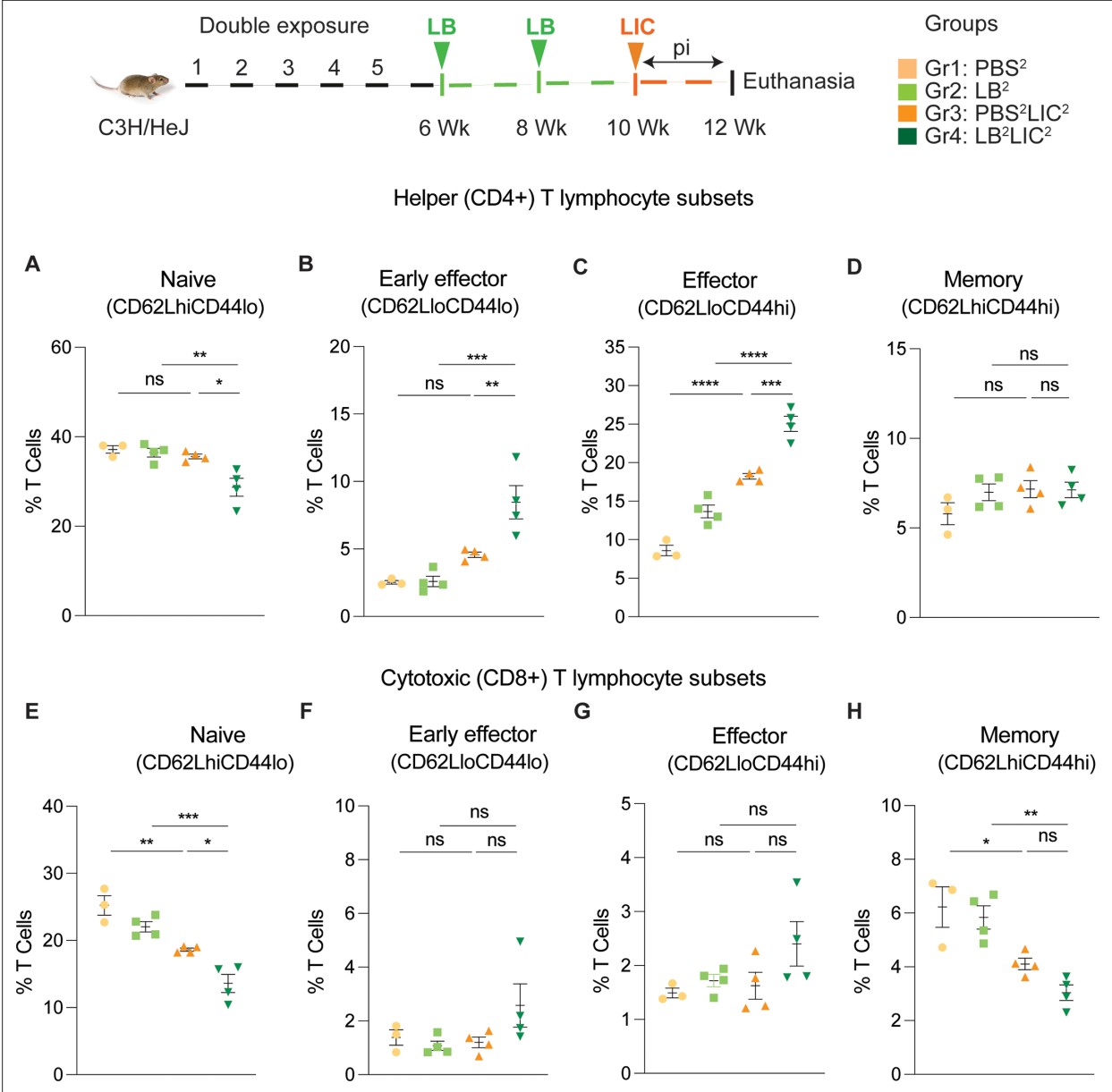

**Figure 6.** Frequency of T cell subsets (CD62L/CD44) in spleen of mice subjected to a double exposure of *L. biflexa* before challenge with *L. interrogans*. (A–D) represent naïve, early effector, effector, and memory subsets of CD4+ helper T lymphocytes, respectively. (E–H) represent naive, early effector, effector, and memory subsets of CD8+ cytotoxic T lymphocytes, respectively. Ordinary one-way ANOVA followed by Tukey's multiple comparison correction test was used to compare between challenged groups and their respective controls; *p<0.05, **p<0.01, ***p<0.001, ****p<0.0001, and ns = not significant; N=3–4 mice per group. Data represents one of two independent experiments.

The online version of this article includes the following source data for figure 6:

**Source data 1.** Excel file containing the source data used to generate *Figure 6*.

that differences with the LIC infected groups were not significant (*Figure 6H*, CD44+CD62L+) (*Figure 6—source data 1*).

## Discussion

Understanding host immune responses to *Leptospira* infection is crucial for advancing our ability to develop new control measures for leptospirosis (*Wunder et al., 2021*; *Vernel-Pauillac et al., 2021*; *Potula et al., 2017*; *Fortes-Gabriel et al., 2022*; *Santecchia et al., 2019*; *Murray et al., 2018*). Given

their widespread presence in nature, the likelihood of humans or animals getting exposed to non-pathogenic serovars of *Leptospira* is likely high (*Benacer et al., 2013*; *Ko et al., 2009*). Our previous studies showed innate immunity engagement during saprophytic *L. biflexa* infection in mice (*Shetty et al., 2021*; *Kundu et al., 2022*). In addition, *L. biflexa* extracts can be used to detect *Leptospira*-specific antibody in up to 67% of serum from patients with clinically confirmed leptospirosis (*Fortes-Gabriel et al., 2022*) which points to a high degree of immunodominant cross-reactive epitopes between *L. biflexa* and pathogenic *Leptospira*. Further, the current hypothesis regarding evolution of *Leptospira* species is that symbiosis of *Leptospira* with eukaryotes emerged from free-living ancestral species (*Thibeaux et al., 2018*); in other words, pathogenic *Leptospira* may have evolved from an environmental ancestor (*Vincent et al., 2019*). Thus, we hypothesized that these highly cross-reactive immunodominant epitopes may also induce cross-protective immune responses. The objective of the current study was to assess whether exposure to a live saprophytic serovar of *Leptospira* provides any heterologous cross-species protection against a subsequent challenge with a pathogenic serovar in a mammalian host (mouse).

In the initial analysis of pathogenesis (*Figure 1*, *Figure 2*) we observed that prior exposure to one or two doses of saprophytic *L. biflexa* rescues weight loss in mice challenged with pathogenic *L. interrogans* at 8 weeks (LB[1]LIC[1]) and at 10 weeks (LB[2]LIC[2]), respectively. Weight gain correlated with survival (75% survival in LB[1]LIC[1] vs 0% survival of the PBS[1]LIC[1] group) in the single exposure experiment, where we expected all mice infected at 8 weeks with pathogenic *L. interrogans* (PBS[1]LIC[1]) to irreversibly lose weight and meet endpoint criteria for euthanasia before the 2-week term of the experiment (*Fortes-Gabriel et al., 2022*, *Shetty et al., 2022*). Loss of mice due to irreversible weight loss is not expected if mice are infected with LIC at 10 weeks of age (*Nair et al., 2020*), as observed in the *L. interrogans* control group in the double exposure experiment (PBS[2]LIC[2]). In both experiments, mice exposed to *L. biflexa* before challenge with LIC produced evidence of *L. interrogans* dissemination in blood, shedding in urine, and kidney colonization.

Histological inspection of kidney slices (*Figure 3*) showed that exposure to a saprophytic *Leptospira* before challenge supported normal structural morphology and prevented infiltration of immune cells in both single and double exposure experiments; in addition, it significantly reduced a fibrosis marker (ColA1) in the single exposure experiment. In the double exposure experiment, differences in ColA1 fibrosis marker are not significant between the two LIC infected groups because 10-week-old C3H-HeJ infected with LIC are more resistant to pathology resultant from infection. Our findings are intriguing as they suggest that while prior live saprophytic exposure did not prevent infection or leptospiral dissemination, it may confer protection against kidney fibrosis.

Our data also shows that prior exposure to non-pathogenic *Leptospira* before pathogenic challenge induced higher antibody titer in the serum, specifically IgG2a antibodies against *L. interrogans* in both single and double exposure experiments (*Figure 4*). Increased IgG2a response in serum is associated with induction of a Th1 biased immune response. Others have recently found that saprophytic *L. biflexa* induced Th1 responses, higher T cell proliferation, and IFN-γ producing CD4+ T cells (*Krangvichian et al., 2023*). Persistent IgM and strain-specific IgG responses were observed during a homologous leptospiral challenge in C57BL6/J mice (*Vernel-Pauillac et al., 2021*). In our study, exposure to a saprophytic *Leptospira* induced antibody responses that may provide heterologous protection against the pathogenic strain of *Leptospira*. This supports a promising broad-spectrum efficacy. Thus, live vaccines derived from a saprophytic strain of *Leptospira* could offer broader protection and overcome the limitation of serovar specificity often observed with killed whole-cell vaccines based on pathogenic strains.

Differences in antibody titer among the *L. interrogans* infected group pre-exposed to saprophytic *L. biflexa* can be attributed to the robust trafficking and differentiation of B and helper T cells (CD4+) measured in spleen (*Figure 5* and *Figure 6*). Presence of effector helper T (CD4+) cells in the spleen indicate a robust cellular immune response as these cells produce cytokines that play a pivotal role in activating other immune cells, including antibody-producing B cells. Moreover, our findings align with another observation which further reinforces the potential immunostimulatory properties of components (polar lipids) derived from saprophytic *L. biflexa*, indicating that these components could play a crucial role in inducing robust B cell responses (*Faisal et al., 2009*). Induction of helper T cell responses along with dynamic transition from naïve to early effector and effector without T helper memory reflects an orchestrated immune response upon pathogenic challenge in the saprophytic

pre-exposed group that is typical of effective responses to vaccines. Previous studies have highlighted the significance of activated CD4+ T cells during *Leptospira* infection in providing protective immunity to the host and mitigating the severity of leptospirosis by releasing cytokines (*Volz et al., 2015*). Correlating induction of chemo-cytokines by saprophytic *Leptospira* with subsequent adaptive immune responses, such as the activation of CD4+ T cells or the production of specific antibodies, provides insights into how innate immune signals drive the adaptive immune response against a pathogenic threat. It may also aid in identifying key signaling molecules or pathways that could be targeted for therapeutic interventions or vaccine design.

While other researchers have explored vaccination strategies using live attenuated or mutant strains of pathogenic serovars, our approach was to utilize a live saprophytic bacterial strain which is unique in the field (*Wunder et al., 2021*; *Potula et al., 2017*; *Lauretti-Ferreira et al., 2020*; *Teixeira et al., 2019*). We previously showed that oral delivery of a probiotic strain, *Lactobacillus plantarum*, reduces the severity of leptospirosis by recruiting myeloid cells (*Fortes-Gabriel et al., 2022*) which suggests that a general phenomenon of trained immunity may be involved. Current vaccines based on inactivated pathogenic species provide equivalent protection to the one achieved in this study (*Barazzone et al., 2021*; *Vernel-Pauillac and Werts, 2018*) with the caveat of being serovar-specific (*Teixeira et al., 2019*; *Vernel-Pauillac and Werts, 2018*; *Adler, 2015*). Although our current study conclusively shows protection from severe leptospirosis after heterologous challenge, it remains to be shown if protection extends to multiple pathogenic serovars of *Leptospira*. Using a live saprophytic strain of *Leptospira* as control strategy could pave the way for development of novel broadly effective vaccines against leptospirosis. Such a vaccine could have a substantial economic impact if applied to animals of agricultural interest.

Another interesting aspect of our current study is that it shows that exposure to a live saprophytic strain of *Leptospira* provides protection against a pathogenic serovar. Thus, in the real-life scenario where individuals or animals may naturally encounter a saprophytic *Leptospira* species, they may develop immune responses that mitigate severe disease outcomes if the host later encounters a pathogenic strain of *Leptospira*. By exploring the immune dynamics during the co-exposure to different *Leptospira* serovars, this study could open avenues of research on strategies that leverage natural exposure to saprophytic species to devise safe control measures against leptospirosis. This concept is important for understanding the epidemiological risk factors of leptospirosis and it should be applicable to other infectious diseases caused by direct contact between the pathogen and mucosal membranes or abraded host skin.

Importantly, we found that in mice pre-exposed to live saprophytic *Leptospira*, there was a correlation between kidney health after LIC infection (less infiltration of immune cells in kidney and less fibrosis marker ColA1) and higher shedding of live LIC in urine. This suggest that a status of homeostasis was reached after kidney colonization that helps the spirochete complete its enzootic cycle. Additional research is needed to fully understand the mechanisms involved in kidney homeostasis after LIC infection.

## Materials and methods

### Animals

Male C3H/HeJ mice (n=6–8/group) were purchased from The Jackson Laboratory (Bar Harbor, ME, USA) and were maintained in a pathogen-free environment in the Laboratory Animal Care Unit of the University of Tennessee Health Science Center (UTHSC). All experiments were performed in compliance with the UTHSC Institutional Animal Care and Use Committee (IACUC), Protocol no. 19-0062.

### Bacteria

Non-pathogenic *L. biflexa* serovar Patoc (LB) belonging to subclade S1 was purchased from ATCC and grown in EMJH media. Pathogenic *L. interrogans* serovar Copenhageni strain Fiocruz L1-130 (LIC) belonging to subclade P1+ (high-virulence pathogens) (*Vincent et al., 2019*; *Giraud-Gatineau et al., 2024*) was grown in EMJH media and subsequently passaged in hamster to maintain virulence. EMJH culture passage 2 was used to inoculate mice ($10^8$) after counting *Leptospira* under a dark-field microscope (Zeiss USA, Hawthorne, NY, USA) using a Petroff-Hausser chamber.

## Infection of mice and study design

We carried out two experiments set apart by a single or double exposure to a saprophytic serovar of *Leptospira* (*L. biflexa*) before challenge with a pathogenic serovar (*L. interrogans*). Groups of mice were inoculated with $10^8$ *Leptospira* intraperitoneally (IP) both for exposure to *L. biflexa* and for challenge with *L. interrogans*. Each experiment was reproduced once. In the single exposure study (*Figure 1A*), Group 1 (n=3) was the naive control which received PBS (PBS[1]), Group 2 (n=4) was inoculated with $10^8$ *L. biflexa* once at 6 weeks (LB[1]), Group 3 (n=4) received PBS for 2 weeks followed by challenge with $10^8$ *L. interrogans* at 8 weeks (PBS[1]LIC[1]), and Group 4 (n=4) was inoculated with $10^8$ *L. biflexa* at 6 weeks and challenged with $10^8$ *L. interrogans* at 8 weeks (LB[1]LIC[1]). In the double exposure study (*Figure 2A*), Group 1 (n=3) was the naive control which received PBS (PBS[2]), Group 2 (n=4) received $10^8$ *L. biflexa* IP at 6 and 8 weeks (LB[2]), Group 3 (n=4) received PBS for 2 weeks followed by challenge with $10^8$ *L. interrogans* at 10 weeks (PBS[2]LIC[2]), and Group 4 (n=4) was inoculated with $10^8$ *L. biflexa* at 6 and 8 weeks and challenged with $10^8$ *L. interrogans* at 10 weeks (LB[2]LIC[2]). Weight was monitored daily. Mice were euthanized 15 days after *L. interrogans* challenge or when they reached the endpoint criteria (20% body weight loss post infection). Blood and kidney were collected at euthanasia: blood was used for quantification of anti-*Leptospira* antibody; kidney was used for quantification of *Leptospira* load (16S rRNA) and it was cultured in EMJH media for evaluation of bacterial viability. Furthermore, kidney samples were stored in 10% formalin for H&E staining. Spleen for flow cytometric analysis was collected from mice after euthanasia only in the double exposure study given that all mice consistently survived challenge.

## *Leptospira* detection through qPCR

Isolation of genomic DNA from blood, urine, and kidney were carried out using NucleoSpin tissue kit (Clontech, Mountain View, CA, USA) according to the manufacturer's instructions. *Leptospira* 16S rRNA primers (Forward- CCCGCGTCCGATTAG and Reverse- TCCATTGTGGCCGAACAC) and TAMRA probe (CTCACCAAGGCGACGATCGGTAGC) were used for detection of *Leptospira* genus using qPCR with a standard curve of $10^5$ to 1 *L. interrogans* (*Nair et al., 2020*, *Richer et al., 2015*). Similarly, qPCR was performed with kidney tissues placed in EMJH to grow live *Leptospira* after culturing for 5 days and visually quantified under a dark-field microscope (20×, Zeiss USA, Hawthorne) on d3 and d5 post culture inoculation.

## RNA isolation and RT-PCR

Kidneys were stored in RNA later after euthanasia. RNeasy Mini Kit (QIAGEN) was used to extract total RNA from kidney tissue according to the manufacturer's specifications. RNA purity was measured using a Nanodrop instrument (Thermo Scientific) at A260/280 ratio. cDNA was prepared using cDNA reverse transcription kit (Applied Biosystems). ColA1 primers (Forward- TAAGGGTACCGCTGGA GAAC, Reverse- GTTCACCTCTCTCACCAGCA), TAMRA probe (AGAGCGAGGCCTTCCCGGAC ), and β-actin primers (Forward- CCACAGCTGAGAGGGAAATC, Reverse- CCAATAGTGATGACCT GGCCG), TAMRA probe (GGAGATGGCCACTGCCGCATC) were purchased from Eurofins Genomics.

## Histopathology by H&E staining

Kidney tissues were fixed in formalin buffer. Histopathology was performed at the Histology Department, UT Methodist University Hospital, Memphis, TN. Digital scanning of inflammatory cell infiltration was measured by taking images of ~5 fields per sample under ×20 magnification. Images were captured after digitally scanning the H&E slides using Panoramic 350 Flash III (3D Histech, Hungary) and CaseViewer software.

## ELISA

Leptospiral extract for *L. biflexa* and *L. interrogans* were prepared as described previously (*Fortes-Gabriel et al., 2022*). Briefly, *Leptospira* was cultured in EMJH media and once confluency was observed, cells were centrifuged to obtain a pellet. This pellet was then incubated with BugBuster solution (1 mL) at room temperature (RT) in a shaker incubator (100 rpm) for 20 min and homogenized by vortexing. Stocks were stored at –20°C. This whole-cell extract of *Leptospira* was then diluted in 1× sodium carbonate coating buffer. Nunc MaxiSorp flat-bottom 96-well plates (eBioscience, San Diego, CA, USA) were coated with extracts prepared from $10^7$ to $10^8$ bacteria per well and incubated

at 4°C overnight. Cells were washed using 1× PBST the following day and blocked for 1 hr using 1% BSA solution, followed by another wash with 1× PBST. Serum samples (1:100) were added to the antigen-coated wells and incubated at 37°C for 1 hr, washed twice with 1× PBST, followed by HRP conjugate secondary anti-mouse- IgG1, IgG2a, and IgG3 (1:10,000) which was incubated for 30 min. After washing the plate three times with 1× PBST the color was developed using TMB SureBlue followed by Stop solution before the absorbance was measured at OD 450 nm using an ELISA plate reader (Molecular Devices Spetramax).

## Flow cytometry

Spleens were chopped into small pieces and macerated to prepare single-cell suspensions on the same day of euthanasia to avoid loss of cell viability. RBC lysis was performed using ACK lysis buffer (Gibco). AO/PI dual staining was used to count live/dead cells on a Luna counter (Logos Biosystems, South Korea). $10^6$ cells were seeded in a 96-well microtiter plate after washing with 1× PBS twice. Blocking was performed with anti-mouse CD16/32 antibody (1:100), followed by 20 min incubation on ice. Fluorochrome-conjugated antibodies (*Supplementary file 1*) were used to stain specific cell surface markers after 30 min incubation in the dark at 4°C. Freshly prepared flow staining buffer was used for washing stained cells. Cells were fixed using 4% paraformaldehyde for 10–15 min at room temperature. Beads were stained using specific fluorochrome-conjugated antibodies and used for compensation, while FMO prepared with spleen cells simultaneously were used for gating controls. Cells were resuspended in flow staining buffer and the Bio-Rad ZE5 cell analyzer was used for data acquisition. Data analysis was done using FlowJo software.

## Statistical analysis

One-way ANOVA with Tukey's multiple comparison test and unpaired t-test with Welch's correction were used to analyze differences between experimental groups. GraphPad Prism software was used to plot graphs; a value of p<0.05 is considered significant. p-Values from all figures for the different experimental groups analyzed by one-way ANOVA and compared with Tukey's multiple comparison test are provided in *Supplementary file 2*.

## Acknowledgements

We sincerely thank Dr. Diedre Daria and Dr. Tony Marion of the Flow Cytometry and Cell Sorting Core facility at UTHSC. We would like to acknowledge the Histology department from UT Methodist University Hospital for processing and staining tissues, Michelle Morrison from the Department of Pathology, UTHSC for digitally scanning tissue slides for histology analysis. This work was supported by the National Institute of Allergy and Infectious Diseases (NIAID), United States National Institutes of Health (NIH), grant numbers R01 AI139267 (MGS), R21 AI 142129 (MGS). The content of this manuscript is totally the responsibility of the authors and does not involve the official views of NIAID or NIH.

## Additional information

### Competing interests

Suman Kundu, Maria Gomes-Solecki: has a provisional patent application (application number 63/618,708) with the United States Patent and Trademark Office (USPTO). The other author declares that no competing interests exist.

### Funding

| Funder | Grant reference number | Author |
|---|---|---|
| National Institute of Allergy and Infectious Diseases | R01 AI139267 | Suman Kundu Maria Gomes-Solecki |
| National Institute of Allergy and Infectious Diseases | R21 AI142129 | Suman Kundu Advait Shetty Maria Gomes-Solecki |

| Funder | Grant reference number | Author |
|---|---|---|

The funders had no role in study design, data collection and interpretation, or the decision to submit the work for publication.

## Author contributions

Suman Kundu, Conceptualization, Formal analysis, Investigation, Methodology, Writing – original draft; Advait Shetty, Investigation, Methodology; Maria Gomes-Solecki, Conceptualization, Formal analysis, Supervision, Funding acquisition, Writing – original draft, Project administration, Writing – review and editing

## Author ORCIDs

Suman Kundu https://orcid.org/0000-0002-8591-1166
Advait Shetty https://orcid.org/0000-0002-4843-0216
Maria Gomes-Solecki https://orcid.org/0000-0002-3715-4543

## Ethics

This study was performed in strict accordance with the recommendations in the Guide for the Care and Use of Laboratory Animals of the National Institutes of Health. All of the animals were handled according to approved institutional animal care and use committee (IACUC) protocol (#19-0062) of the University of Tennessee Health Science Center (UTHSC).

Reviewer #3 (Public Review): https://doi.org/10.7554/eLife.96470.3.sa1
Author response https://doi.org/10.7554/eLife.96470.3.sa2

# Additional files

## Supplementary files

• Supplementary file 1. Table includes the list of primary fluorochrome-conjugated antibodies used in flow cytometry staining for spleen immune phenotyping.

• Supplementary file 2. Table includes the p-values from all the figures for the different experimental groups analyzed by one-way ANOVA and compared with Tukey's multiple comparison test.

• MDAR checklist

## Data availability

All data generated or analyzed during this study are included in this manuscript and source data files.

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
