## [Editor Report · eLife assessment]

This **important** study contributes to our understanding on how prior exposure to a non-pathogenic Leptospira strain could prime the host to prevent severe leptospirosis following infection with a pathogenic strain. The work described is **solid** and broadly supports the claims, with minor weaknesses that could be addressed in future studies. The work will be of interest to scientists interested in host-pathogen interactions and leptospirosis.

---

## [Referee Report · Reviewer #3 (Public Review)]

Summary:

Kundu et al. investigated the effects of pre-exposure to a non-pathogenic Leptospira strain in prevention of severe disease following subsequent infection by a pathogenic strain. They utilized a single or double exposure method to the non-pathogen prior to challenge with a pathogenic strain. They found that prior exposure to a non-pathogen prevented many of the disease manifestations of the pathogen. Bacteria, however, were able to disseminate, colonize the kidneys, and be shed in the urine. This is important foundational work to describe a novel method of vaccination against leptospirosis. Numerous studies have attempted to use recombinant proteins to vaccinate against leptospirosis, with limited success. The authors provide a new approach that takes advantage of the homology between a non-pathogen and a pathogen to provide heterologous protection. This will provide a new direction in which we can approach creating vaccines against this re-emerging disease.

Strengths:

The major strength of this paper is that it is one of the first studies utilizing a live non-pathogenic strain of Leptospira to immunize against severe disease associated with leptospirosis. They utilize two independent experiments (a single and double vaccination) to define this strategy. This represents a very interesting and novel approach to vaccine development. This is of clear importance to the field.

The authors use a variety of experiments to show the protection imparted by pre-exposure to the non-pathogen. They look at disease manifestations such as death and weight loss. They define the ability of Leptospira to disseminate and colonize the kidney. They show the effects infection has on kidney architecture and a marker of fibrosis. And they begin to define the immune response in both of these exposure methods. This provides evidence of the numerous advantages this vaccination strategy may have. Thus, this study provides an important foundation for future studies utilizing this method to protect against leptospirosis.

Weaknesses:

A direct comparison between single and double exposure to the non-pathogen is not possible with the data presented. The ages of mice infected were different between the single (8 weeks) and double (10 weeks) exposure methods, thus the phenotypes associated with LIC infection are different at these two ages. The authors state that this is expected, but do not provide a reasoning for this drastic difference in phenotypes. It cannot be determined if double-vaccination would provide an additional benefit, which is of importance to future work developing any vaccine treatment. An experiment directly comparing the two exposure methods while infecting mice at the same age would be of great relevance to and strengthen this work.

---

## [Author Response]

The following is the authors’ response to the original reviews.

**Public Reviews:**

**Reviewer #1 (Public Review):**
As a reviewer for this manuscript, I recognize its significant contribution to understanding the immune response to saprophytic Leptospira exposure and its implications for leptospirosis prevention strategies. The study is well-conceived, addressing an innovative hypothesis with potentially high impact. However, to fully realize its contribution to the field, the manuscript would benefit greatly from a more detailed elucidation of immune mechanisms at play, including specific cytokine profiles, antigen specificity of the antibody responses, and long-term immunity. Additionally, expanding on the methodological details, such as immunophenotyping panels, qPCR normalization methods, and the rationale behind animal model choice, would enhance the manuscript's clarity and reproducibility. Implementing functional assays to characterize effector T-cell responses and possibly investigating the microbiota's role could offer novel insights into the protective immunity mechanisms. These revisions would not only bolster the current findings but also provide a more comprehensive understanding of the potential for saprophytic Leptospira exposure in leptospirosis vaccine development. Given these considerations, I believe that after substantial revisions, this manuscript could represent a valuable addition to the literature and potentially inform future research and vaccine strategy development in the field of infectious diseases.
**Reviewer #2 (Public Review):**
Summary:The authors try to achieve a method of protection against pathogenic strains using saprophytic species. It is undeniable that the saprophytic species, despite not causing the disease, activates an immune response. However, based on these results, using the saprophytic species does not significantly impact the animal's infection by a virulent species.Strengths:Exposure to the saprophytic strain before the virulent strain reduces animal weight loss, reduces tissue kidney damage, and increases cellular response in mice.Weaknesses:Even after the challenge with the saprophyte strain, kidney colonization and the release of bacteria through urine continue. Moreover, the authors need to determine the impact on survival if the experiment ends on the 15th.
**Reviewer #3 (Public Review):**
Summary:Kundu et al. investigated the effects of pre-exposure to a non-pathogenic Leptospira strain in the prevention of severe disease following subsequent infection by a pathogenic strain. They utilized a single or double exposure method to the non-pathogen prior to challenge with a pathogenic strain. They found that prior exposure to a non-pathogen prevented many of the disease manifestations of the pathogen. Bacteria, however, were able to disseminate, colonize the kidneys, and be shed in the urine. This is an important foundational work to describe a novel method of vaccination against leptospirosis. Numerous studies have attempted to use recombinant proteins to vaccinate against leptospirosis, with limited success. The authors provide a new approach that takes advantage of the homology between a non-pathogen and a pathogen to provide heterologous protection. This will provide a new direction in which we can approach creating vaccines against this re-emerging disease.Strengths:The major strength of this paper is that it is one of the first studies utilizing a live non-pathogenic strain of Leptospira to immunize against severe disease associated with leptospirosis. They utilize two independent experiments (a single and double vaccination) to define this strategy. This represents a very interesting and novel approach to vaccine development. This is of clear importance to the field.The authors use a variety of experiments to show the protection imparted by pre-exposure to the non-pathogen. They look at disease manifestations such as death and weight loss. They define the ability of Leptospira to disseminate and colonize the kidney. They show the effects infection has on kidney architecture and a marker of fibrosis. They also begin to define the immune response in both of these exposure methods. This provides evidence of the numerous advantages this vaccination strategy may have. Thus, this study provides an important foundation for future studies utilizing this method to protect against leptospirosis.Weaknesses:Although they provide some evidence of the utility of pretreatment with a non-pathogen, there are some areas in which the paper needs to be clarified and expanded.The authors draw their conclusions based on the data presented. However, they state the graphs only represent one of two independent experiments. Each experiment utilized 3-4 mice per group. In order to be confident in the conclusions, a power analysis needs to be done to show that there is sufficient power with 3-4 mice per group. In addition, it would be important to show both experiments in one graph which would inherently increase the power by doubling the group size, while also providing evidence that this is a reproducible phenotype between experiments. Overall, this weakens the strength of the conclusions drawn and would require additional statistical analysis or additional replicates to provide confidence in these conclusions.A direct comparison between single and double exposure to the non-pathogen is not able to be determined. The ages of mice infected were different between the single (8 weeks) and double (10 weeks) exposure methods, thus the phenotypes associated with LIC infection are different at these two ages. The authors state that this is expected, but do not provide a reasoning for this drastic difference in phenotypes. It is therefore difficult to compare the two exposure methods, and thus determine if one approach provides advantages over the other. An experiment directly comparing the two exposure methods while infecting mice at the same age would be of great relevance to and strengthen this work.
**Recommendations for the authors:**

**Reviewer #1 (Recommendations For The Authors):**
Major Comments(1) Elucidation of Immune Mechanisms: The manuscript intriguingly suggests that exposure to saprophytic Leptospira primes the host for a Th1-biased immune response, contributing to survival and mitigation of disease severity upon subsequent pathogenic challenge. However, the underlying mechanisms remain broadly defined. A more detailed investigation into the cytokine profiles, particularly the levels of IFN-γ, IL-12, and other Th1-associated cytokines, could clarify the mechanism of Th1 bias. Moreover, exploring the role of antigen-presenting cells (APCs) in priming T cells towards a Th1 phenotype would add valuable insights.

In this study we continue to elucidate the immune mechanisms engaged by pathogenic and non-pathogenic Leptospira as a follow up to our previous work (Shetty et al, 2021 PMID: 34249775, and Kundu et al 2022 PMID 35392072). We, and others, have shown that saprophytic *L. biflexa* and pathogenic *L. interrogans* induce major chemo-cytokines associated with Th1 biased immune responses (Shetty et al. 2021; Cagliero et al. 2022; Krangvichian et al. 2023) and engage myeloid immune cells such as macrophages and dendritic cells. The role of antigen presenting cells such as dendritic cells in priming T cells and activating adaptive response is a separate question and can be addressed in the future. To further address this question, a recent mechanistic study (Krangvichian et al. 2023) showed that non-pathogenic leptospires (*L. biflexa*) promote MoDC maturation and stimulate the proliferation of IFN-γ-producing CD4+ T cells and potentially elicit a Th1-type response in mice, which also supports our current claim and it is referenced in our manuscript.

(2) Quantitative Analysis of Kidney Colonization: The manuscript reports that pre-exposure to L. biflexa did not prevent the colonization of kidneys by L. interrogans but led to a more regulated immune response and reduced fibrosis. A more nuanced quantification of bacterial loads in the kidneys, using techniques such as CFU counting or more sensitive qPCR methods, could provide a clearer picture of how saprophytic exposure affects the ability of pathogenic Leptospira to establish infection. Additionally, a time-course study showing the kinetics of bacterial colonization and clearance post-infection would be informative.

We are currently validating digital PCR to use in the future and plan to do time course studies.

(3) Characterization of B Cell and T Cell Responses: While the manuscript mentions increased B cell frequencies and effector T helper cell responses, specifics regarding the nature of these responses are lacking. For instance, detailing the isotype and specificity of antibodies produced, the proliferation rates of specific B and T cell subsets, and their functional capabilities (e.g., cytotoxicity, help for B cells) would significantly enrich the understanding of the immune response elicited by pre-exposure to saprophytic Leptospira.

Indeed, additional experiments need to be conducted to flush out the immune responses engaged after pre-exposure to saprophytic *Leptospira* followed by LIC challenge.

(4) Comparative Analysis with Other Models of Pre-exposure: The study primarily focuses on pre-exposure to a live saprophytic Leptospira. Including a comparison with pre-exposure to killed saprophytic bacteria, or even to other non-pathogenic microbes, could help discern whether the observed protective effect is unique to live saprophytic Leptospira exposure or if it represents a more general phenomenon of trained immunity.

Regarding the use of other non-pathogenic microbes, our lab has shown in the past that oral use of probiotic strain *Lactobacillus plantarum* (Potula et al 2017) also reduces the severity of Leptospirosis by recruiting myeloid cells. Thus, there may be a general phenomenon of trained immunity involved. We added this to the discussion.

(5) Assessment of Long-term Immunity: The study provides valuable insights into the short-term outcomes following saprophytic Leptospira exposure and subsequent pathogenic challenge. Extending these observations to assess long-term immunity, including memory B and T cell responses several months post-infection, would be crucial for understanding the potential of saprophytic Leptospira exposure in providing lasting protection against leptospirosis.

Long term immunity is a complex and separate question that we plan to address later.

Minor Comments(1) Technical Specifics of Flow Cytometry Analysis: The manuscript could benefit from including more details on the flow cytometry gating strategy and the specific markers used to identify different immune cell subsets. This addition would aid in the reproducibility of the results and allow for a clearer interpretation of the immune profiling data.

We included the technical specifics of the flow-cytometry analysis in the materials and methods section. The gating strategy (Fig S1) and the specific markers (TableS1) used to identify different immune cell subsets were incorporated in the supplementary datasheet. The cell specific markers were incorporated in the figures (Fig 5 and 6) under each representative cell subset which facilitates clarity and reproducibility of immune profiling.

(2) Statistical Methodology for IgG Subtyping: The analysis of IgG subtypes in response to Leptospira exposure is intriguing but would be strengthened by specifying the statistical tests used to compare IgG1, IgG2a, and IgG3 levels between groups. Additionally, discussing the biological significance of the observed differences in IgG subtype levels would provide a more comprehensive understanding of the immune response.

We applied the ordinary One-way ANOVA test to compare the IgG subtypes between groups followed by a Tukey’s multiple comparison correction analysis (included in the figure legend of Fig 4). We addressed the biological relevance of the observed differences in IgG subtype levels in the discussion section.

(3) Details on Animal Welfare and Ethical Approval: While the manuscript mentions compliance with institutional animal care and use committee protocols, providing the specific ethical guidelines followed, such as the 3Rs (Replacement, Reduction, Refinement), would reinforce the commitment to ethical research practices.

This is addressed in our institutional IACUC which is approved and listed in Methods.

(4) Clarification of Figure Legends: Some figure legends are brief and could be expanded to more thoroughly describe what the figures show, including details on what specific data points, error bars, and statistical symbols represent.

We updated and expanded the figure legends (Fig 1-4).

(5) Revision of Introduction and Background: The introduction provides a good overview of leptospirosis and the rationale behind the study. However, it could be further improved by briefly summarizing current challenges in vaccine development against leptospirosis and how understanding the immune response to saprophytic Leptospira could address these challenges.

We revised the introduction keeping this comment in mind.

**Reviewer #2 (Recommendations For The Authors):**
- Perform the same challenge experiment with a hamster.

We clarified throughout the manuscript that all the work was done using the C3H-HeJ mouse model which was developed in our lab for the purpose of measuring differences in sublethal and lethal LIC infections. We leave the experiments using hamster to the investigators that have thoroughly validated the hamster model of lethal Leptospira infection.

- Review the written part where it is understood that the challenge with saprophyte strain before virulence prevents the disease.

We reviewed the manuscript to be understood that inoculation of mice with a saprophyte *Leptospira* before pathogenic challenge prevents severe leptospirosis and promotes kidney homeostasis and increased shedding of *Leptospira* in urine which is interesting. The last 2 sentences of the abstract read: “Thus, mice exposed to live saprophytic *Leptospira* before facing a pathogenic serovar may withstand infection with far better outcomes. Furthermore, a status of homeostasis may have been reached after kidney colonization that helps LIC complete its enzootic cycle.”

**Reviewer #3 (Recommendations For The Authors):**
(1) Line 83: The authors refer to the classification of Leptospira by old nomenclature. The bacteria are now categorized into clades P1, P2, S1 and S2. See Vincent et al. Revisiting the taxonomy and evolution of pathogenicity of the genus Leptospira through the prism of genomics. PLoS Negl Trop Dis. 2019 May 23;13(5):e0007270. doi: 10.1371/journal.pntd.0007270. PMID: 31120895; PMCID: PMC6532842.

We have included the categories (S1 for *L. biflexa* and P1+ for *L. interrogans*) in introduction and methods but we did not update the figures because we want to be specific about the species used in these experiments. We also include a few sentences on evolution of *Leptospira* species in discussion and reference Thibeaux 2018, Vincent 2019 and Giraud-Gatineau, 2024.

(2) Line 133: Please remove the extra line to be consistent with the rest of the method section format.

We addressed all formatting issues.

(3) Line 137: Are these primers specific to pathogenic L. interrogans? Or do they cross react with L. biflexa? If not specific, how long does L. biflexa stick around after infection?

The primers are specific to the genus *Leptospira*. Surdel et al. in 2022 used 16s rRNA target sequence to amplify *L. biflexa* Patoc in mice at 6 hours post infection. We did not detect any positive sample for *L. biflexa* with the 16s rRNA primer set because we do our analysis 30 days and 45 days post inoculation with *L. biflexa.* We clarified this issue in methods and results.

(4) Statistical analysis:(a) Some of your graphs have more than 4 points on them (such as Figure 4), while the legend still reads "represents one of two independent experiments". Are these actually combined replicates in the same graph? Combining them would provide strength to your conclusions throughout your manuscript and may provide stronger power for comparisons. If they are not included, why are they not included together? Please clarify what is included in each graph, and why the two experiments were not included together.

We updated the legends with the total number of mice used in the experiment represented in the figure. Figures 1, 2, 4 and S2 contain the combined results from two independent experiments. Figures 3, 5 and 6 represent data from one of two independent experiments. For Fig 3 it would be redundant to show HE images of two experiments. Regarding Figs 5 and 6, the flow-cytometry equipment acquires data at different voltage every single time and biological samples vary between experiments even if all the markers and procedures are the same. So, we reproduce the experiment and show results from one experiment after confirming that the trend between individual experiments are the same.

(b) If ANOVA was used, were all columns compared to each other? Why in some graphs are "ns" labeled only for certain comparisons? I would suggest removing the "ns" comparisons and only highlighting the significant differences.

We have incorporated the comparison analysis between control (PBS) versus the PBS-LIC, LB versus LB-LIC and PBS-LIC versus LB-LIC in both the studies although we have compared significance between all groups.

(5) Line 165: Bacteria were not plated, extract was plated. Perhaps you mean "extract corresponding to 107-108 bacteria"?

We addressed it as follows: “Nunc MaxiSorp flat-bottom 96 well plates (eBioscience, San Diego, CA) were coated with extracts prepared from 107-108 bacteria per well and incubated at 4℃ overnight” …

(6) Line 260: The authors claim that "Exposure to non-pathogenic L. biflexa before pathogenic L. interrogans challenge provided a significant immune cell boost with an increase in overall B and helper T cell frequencies..." However, in Figure 5A, the number of B cells in both the PBS2LIC2 and the LB2LIC2 are not significantly different. Thus, the claim is not supported by the evidence provided. It appears that infection with LIC led to similar increases in B cells regardless of pretreatment.

We rephrased that title to reflect the finding that increased differences were measured in effector Helper T cells between PBS2LIC2 and LB2LIC2 (Figs 5D and 6B, 6C) and we re-wrote this section for clarity.

(7) Lines 314-315: The authors claim that it protected against kidney fibrosis, however, the data only supports that only a single exposure to LB reduced levels of a marker associated with kidney fibrosis. Fibrosis was never directly measured.

Indeed, we didn’t do Mason’s Trichrome stain to get supporting data for kidney fibrosis and only measured a fibrosis marker ColA1. We toned down this section: “ …. it may confer protection against kidney fibrosis.”

(8) Line 317: Authors state that pre-exposure induced higher antibodies in serum, however, this was never shown. Only an increase in IgG2a was shown. Please word this statement to make it clear total antibodies were never measured.

We did measure total anti-*Leptospira interrogans* IgM and IgG antibodies. We added the following sentence to description of these results: “In both experiments, total IgM and IgG were significantly increased in PBS-LIC and LB-LIC when compared to the respective controls, but not between PBS-LIC and LB-LIC. Regarding IgG isotypes, IgG1…”

(9) Line 323: The authors state that the exposure "induced antibody responses that provided heterologous protection." There is no evidence that the protection is due to the antibody response in these experiments. In fact, they also showed that it induced increased T cell responses.

We toned down this statement: “In our study, exposure to a saprophytic *Leptospira* induced antibody responses that may provide heterologous protection against the pathogenic strain of *Leptospira*.”

(10) Line 328: The authors us the term "stark difference", however, only slight differences are seen.

We toned down that statement as follows: “Differences in antibody titer among the *L. interrogans* infected….”

(11) Line 490: reword this sentence to provide clarity and easier to read: "inoculated once with 10^8 L. biflexa at 6 weeks and they were challenged with 10^8 L. interrogans SEROVAR Copenhageni FioCruz (LIC) at 8 weeks."

We revised the sentence.

(12) Figure 1 and 2: Quantifying bacteria in culture after infection is not meaningful, as there are numerous factors that can affect the replication in culture after infection, such as how the organ perhaps was cut before placing it in culture. The comparisons in Figure 2E and F therefore are not interpretable. I would suggest presenting this data as Culture Positive or Culture Negative.

We added these data to the figure under DFM (dark field microscopy).

(13) Figure 3A: H&E staining often leads to different qualities of stains. But is there a better image that can be chosen for the PBS1LIC1 that provides a better comparison with the other images chosen? This is not worth repeating the experiment to get one, just make the figure look better if you have one available.

We screened the images again but the one incorporated in the figure3A for PBS1LIC1 is the best.

(14) Figure 3D: I agree that the PBS-LIC treatment is significant, but please include P value, as it looks very similar to the LB-LIC group. The two LIC groups are not significantly different, so the conclusion would be pre-exposure does not mitigate renal fibrosis marker ColA in the double-exposure study.

We included the p-values in this figure. The two LIC groups are significantly different (ColA1) in the single exposure experiment, and the in double exposure we don’t expect to be able to measure ColA1 differences because the mice are older (10 wk) when we do the LIC challenge.